# MAFLD in Obese Children: A Challenging Definition

**DOI:** 10.3390/children8030247

**Published:** 2021-03-23

**Authors:** Anna Di Sessa, Stefano Guarino, Giuseppina Rosaria Umano, Mattia Arenella, Salvatore Alfiero, Gaetano Quaranta, Emanuele Miraglia del Giudice, Pierluigi Marzuillo

**Affiliations:** Department of Woman, Child and of General and Specialized Surgery, Università degli Studi della Campania “Luigi Vanvitelli”, Via Luigi De Crecchio 2, 80138 Napoli, Italy; anna.disessa@libero.it (A.D.S.); stefano.guarino@policliniconapoli.it (S.G.); mattia.re92@gmail.com (M.A.); salalfiero@gmail.com (S.A.); gaetano.quaranta1103@gmail.com (G.Q.); emanuele.miraglia@unicampania.it (E.M.d.G.); pierluigi.marzuillo@gmail.com (P.M.)

**Keywords:** fatty, liver, metabolic, dysfunction, children, obesity

## Abstract

Background: Recently, the new definition of Metabolic (dysfunction) associated fatty liver disease (MAFLD) has gained remarkable scientific interest. We aimed to evaluate the effectiveness of MAFLD definition in selecting obese children at higher cardiovascular risk. Methods: A total of 954 obese children and adolescents was retrospectively enrolled. Clinical, biochemical, and metabolic evaluations were performed. Hepatic steatosis was assessed by liver ultrasound. According to the metabolic status, the population was divided in three groups. Group 1 included obese patients without both non-alcoholic fatty liver disease (NAFLD) and metabolic dysregulation; group 2 included patients with obesity and NAFLD (then encompassing one MAFLD criterion); group 3 included patients with obesity, NAFLD and evidence of metabolic dysregulation (then encompassing more than 1 MAFLD criteria). Results: Patients of Group 3 showed a worse cardiometabolic profile, as also proven by the higher percentage of prediabetes (defined as the presence of impaired fasting glucose or impaired glucose tolerance) compared to other groups (*p* = 0.001). Conclusions: MAFLD criteria in obese children seem to be less accurate in identifying patients having an intrinsic higher cardiometabolic risk. This suggests the need for a more accurate definition in the context of pediatric obesity.

## 1. Introduction

Non-alcoholic fatty liver disease (NAFLD) represents a heterogeneous and progressive condition with a major pathogenic interplay between both metabolic (e.g., obesity, insulin resistance) and genetic (e.g., PNPLA3 and TM6SF2) factors [1]. Parallel to the rising trend in obesity and diabetes, it has been increasingly recognized as the most common chronic liver disease both in adults and children, representing an enormous global health concern due to its cardiometabolic burden [2,3,4]. Over the past 20 years, accumulating evidence has indicated NAFLD as the hepatic manifestation of a systemic metabolic disorder [1,5]. Of concern, several studies have linked NAFLD to cardiometabolic derangements even in childhood [6]. Despite several promising researches investigating pharmacological approaches, to date no drug therapy for NAFLD has been licensed and lifestyle interventions remain the mainstay of this treatment [3].

Given these alarming data reflecting the expanding knowledge in this field, an international consensus in 2020 has proposed “Metabolic (dysfunction) associated fatty liver disease” (MAFLD) as a new name for NAFLD [7]. Diagnosis of MAFLD is based on the radiological evidence of hepatic steatosis and the presence of at least one of the following criteria namely overweight/obesity, type 2 diabetes (T2D), or evidence of metabolic dysregulation (defined as the presence of two or more of these conditions: (1) Waist circumference >95th percentile for age and sex, (2) blood pressure >95 th for age, sex, and height. (3) Triglycerides >150 mg/dL, (4) HDL < 40 mg/dl, (5) prediabetes, (6) homeostasis model assessment-insulin resistance (HOMA-IR) score >2.5, (7) C-reactive protein (CRP) levels >2 mg/L [7].

The proposed new term does not represent a mere semantic revision but underlies the close association with the metabolic milieu [8,9,10,11]. Given this latter relationship, MAFLD seems to be more useful in identifying patients at risk in adult clinical practice [11,12]. To date, there are still few data testing the MAFLD diagnostic criteria in daily clinical practice and the proposed update in nomenclature is currently being discussed [11,13,14,15,16].

In particular, evidence about the impact of MAFLD definition in pediatric setting is currently limited [17,18].

Considering that not all the obese children are metabolically unhealthy and according to new definition will suffer from MAFLD, it could be helpful improve the MAFLD definition in the setting of pediatric obesity to have a tool being able to adequately select patients at higher cardiovascular risk.

We hypothesize that MAFLD definition in obese children is of limited utility in selecting patients at higher cardiovascular risk.

Because of the lack of pediatric evidence on the usefulness of MAFLD definition in this setting, we aimed to test the ability of MAFLD diagnostic criteria in selecting obese children and adolescents at higher cardiovascular risk because of metabolic dysregulation.

## 2. Materials and Methods

We retrospectively enrolled a cohort of 954 obese children and adolescents evaluated at our Obesity Clinic from June 2017 to June 2020. The study was approved by the research ethical committee of University of Campania “Luigi Vanvitelli” and conducted in accordance with the Declaration of Helsinki. Informed written consent was obtained before any procedure.

An accurate physical examination was performed in all the enrolled subjects. Clinical parameters including pubertal stage and blood pressure were obtained as described [19].

After an overnight fast, an intravenous cannula was inserted in the antecubital vein for withdrawal of blood samples for measurement of the main both biochemical and metabolic parameters. Homeostasis model assessment of insulin-resistance (HOMA-IR) was calculated by the equation fasting insulin (µU/mL) * fasting glucose (mg/dL)/405. Measurements of lipids and other biochemical parameters were obtained by standard laboratory procedures [19].

To assess glucose tolerance, a standard 2-h oral glucose tolerance test (OGTT) was also performed in all the enrolled subjects. Patients assumed 1.75 g of glucose per kilogram of body weight. During OGTT, glucose and insulin levels were measured at 0, 30, 60, 90, and 120 min.

Prediabetes was defined as the presence of impaired fasting glucose (IFG; fasting glucose plasma concentration between 100 mg/dL to <126 mg/dL) or impaired glucose tolerance (IGT; at 2-h plasma glucose concentration after a 75-g oral glucose tolerance test of 140 mg/dL to 199 mg/dL) [19].

Genomic DNA was extracted from peripheral whole blood with a DNA extraction kit (Promega, Madison WI, USA). Patients were also genotyped for the I148M *patatin like phospholipase containing domain 3* (*PNPLA3)* polymorphism by Taqman allelic discrimination assay on ABI 7900HT Real-Time PCR System. Predesigned assay primers and probes were purchased from Applied Biosystems (Foster City, CA, USA) [20].

To detect hepatic steatosis, a trained radiologist performed liver ultrasonography in all the enrolled subjects. The presence or absence of liver steatosis was evaluated using standardized criteria and taking into account the abnormally intense, high-level echoes arising from the hepatic parenchyma and liver–kidney differences in echo amplitude. NAFLD was defined by the presence of ultrasound detected liver steatosis and/or ALT levels >40 IU/L.

### Classification of the Population

The population was divided in 3 groups (Figure 1). Group 1 included obese subjects without both NAFLD and metabolic dysregulation; group 2 included patients with obesity and NAFLD (then encompassing one MAFLD criterion); group 3 included patients with obesity, NAFLD and evidence of metabolic dysregulation (then encompassing more than 1 MAFLD criteria); Metabolic dysregulation was defined as the presence of two or more of these conditions: (1) Waist circumference >95th percentile for age and sex, (2) Blood pressure >95th for age, sex, and height. (3) Triglycerides >150 mg/dL, (4) HDL < 40 mg/dl, (5) prediabetes, (6) homeostasis model assessment-insulin resistance (HOMA-IR) score >2.5, and (7) C-reactive protein (CRP) levels >2 mg/L.

To define the increased cardiovascular risk in our population, we used NAFLD and criteria for metabolic dysregulation because of their close intertwining with cardiovascular disease [1,5,6].

We classified the study population according to the metabolic status and we compared the main clinical and laboratory parameters among these groups.

Student’s t test (for variables normally distributed), Mann–Whitney U test (for variables non-normally distributed), and Chi squared test (for categorical variables) were used to examine the differences among groups. A one-way analysis of variance (ANOVA) was made for comparison among groups.

Continuous variables are shown as mean ± standard deviation. Categorical values are shown as n (%).

The IBM SPSS Statistics software, Version 24 (IBM, Armonk, NY) was used for all statistical analyses. Data were expressed as means ± SD. *p*-values less than 0.05 were considered statistically significant.

## 3. Results

The main clinical and biochemical parameters of the entire Study population and of the cohort stratified into three groups are shown in Table 1. Patients encompassing more than 1 MAFLD criteria (Group 3) were significantly older (*p* = 0.001) and showed higher BMI-SDS (*p* < 0.0001), SBP-SDS (*p* < 0.0001), DBP-SDS (*p* = 0.001), W/Hr (*p* < 0.0001) HOMA-IR (*p* < 0.0001), triglycerides levels (*p* < 0.0001), baseline and 2-h OGTT glycaemia (*p* < 0.0001), and transaminase levels (*p* < 0.0001) than those belonging to other groups. Taken together, these subjects showed a worse cardiometabolic profile, as also proven by the higher percentage of prediabetes (defined as the presence of impaired fasting glucose or impaired glucose tolerance) compared to other groups (*p* = 0.001). Moreover, the patients of Group 3 presented with higher prevalence of carriers of *PNPLA3* rare allele compared with others (*p* = 0.001).

## 4. Discussion

Our findings suggested that the MAFLD diagnosis based on “overweight/obesity” criteria in obese children were less accurate in identifying patients at higher cardiometabolic risk compared with the diagnosis of MAFLD based on “evidence of metabolic dysregulation” and “overweight/obesity” criteria.

In fact, a better characterization of this selected population is provided by the presence of more than one MAFLD criteria, as demonstrated by our preliminary results.

The effectiveness of MAFLD diagnostic criteria in selecting adult patients with fatty liver disease associated with metabolic dysfunction is still debating [12,13,14,21,22].

Noteworthy, as argued by Hegarthy et al., the accuracy of MAFLD definition in emphasizing metabolic dysfunction seems to be less powerful in a highly selected population such as obese young [17]. Nevertheless, a similar potential underestimation of MAFLD criteria has been also recently found in a large adult cohort with severe hepatic steatosis who did not meet MAFLD diagnosis [15].

Although the close association between obesity and liver fat, not all the obese population develop metabolic liver disease and not all the patients with NAFLD develop metabolic dysfunction. In fact, over the recent years two different obesity phenotypes have emerged such as metabolically healthy obesity (MHO) and metabolically unhealthy obesity (MUO), but these definitions and their both clinical and prognostic implications are still under debate [23,24]. Indeed, an adverse cardiometabolic outcome has been observed not only in metabolically unhealthy obese patients but also in subjects classified as MHO phenotype [23]. In this perspective, a timely and adequate stratification of obese children represents a crucial step in an effort to early counteract the cardiometabolic risk of these subjects. Of note, our findings also showed an intriguing role of the *PNPLA3* gene, the most important risk polymorphism for NAFLD, in a more comprehensive metabolic context, as already demonstrated in children [20].

However, this investigation has some limitations that deserve mention. Data regarding CRP levels were available only in a non-significant group of patients. More, hepatic steatosis was diagnosed by liver sonography instead of by biopsy, though better reflecting daily clinical practice, in which this latter is not used for the diagnosis of fatty liver disease. On the other hand, additional tools for NAFLD diagnosis in overweight/obese children have been recently identified [25,26]. In addition to the classical techniques such as computed tomography or magnetic resonance, several attractive approaches in this field have been proposed over the past years [26]. For instance, controlled attenuation parameter and transient elastography represent further noninvasive tools for NAFLD assessment [25,26]. Besides, researchers have also suggested distinct biomarkers (e.g., fetuin A, chemerin, and cathepsin D) or predictive scores (including the Pediatric NAFLD score) for pediatric NAFLD diagnosis [26]. Taken together, these approaches are promising but need to be larger validated.

Given the increasing prevalence of fatty liver and its related adverse outcomes already in childhood, further researches are needed to better elucidate the usefulness of MAFLD diagnostic criteria in the “real pediatric world” in adequately stratifying young patients not only in a general setting but also in a specific context such as obesity having an intrinsic greater cardiometabolic risk.

## Figures and Tables

**Figure 1 children-08-00247-f001:**
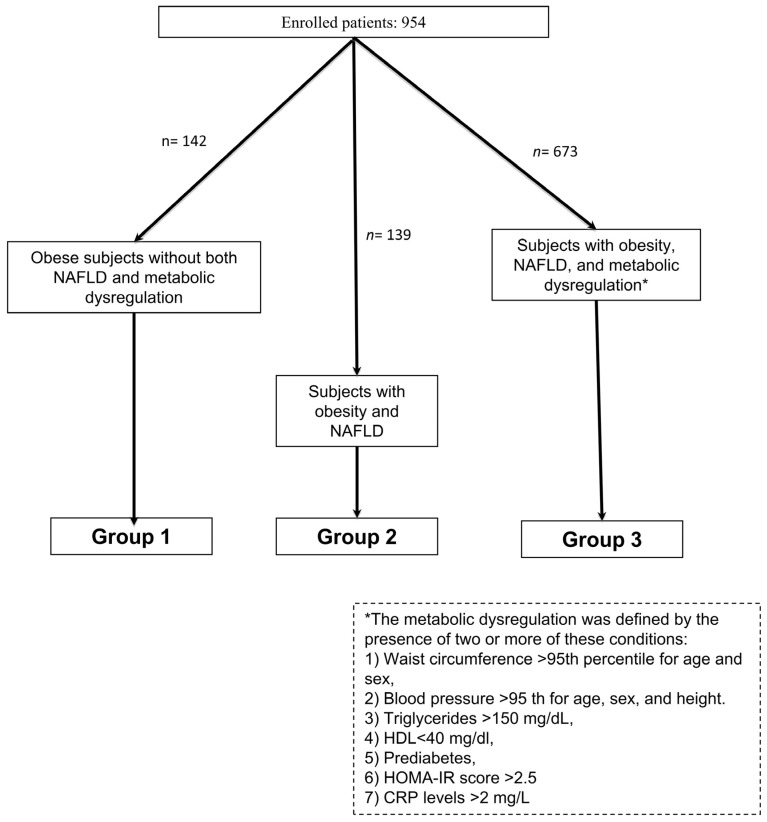
Flow chart summarizing the criteria for the inclusion in the 3 different groups.

**Table 1 children-08-00247-t001:** Main features of the study population according to the metabolic status.

All (*n* = 954)	Group 1 (*n* = 142)	Group 2 (*n* = 139)	Group 3 (*n* = 673)	*p*-Value	*p*1 *	*p*2 **	*p*3 ***
**Age, year** 10.57 ± 2.96	10.18 ± 2.75	10.64 ± 2.86	11.12 ± 2.73	**0.001**	0.08	**0.01**	**0.002**
**BMI–SDS** 2.42 ± 0.57	2.22 ± 0.51	2.46 ± 0.47	2.49 ± 0.57	**<0.0001**	**<0.0001**	**0.001**	**<0.0001**
**Sex (male), %** 52	45.1	62.6	59.3	**<0.0001**	**0.001**	**0.001**	**<0.0001**
**SBP-SDS** 0.53 ± 1.10	0.23 ± 0.88	0.57 ± 1.03	0.68 ± 1.17	**<0.0001**	**<0.0001**	**<0.0001**	**<0.0001**
**DBP-SDS** 0.25 ± 0.72	0.07 ± 0.60	0.25 ± 0.61	0.31 ± 0.76	**0.001**	**0.002**	**0.001**	**<0.0001**
**W/Hr** 0.61 ± 0.06	0.58 ± 0.56	0.60 ± 0.06	0.63 ± 0.05	**<0.0001**	**<0.0001**	**0.001**	**<0.0001**
**Waist (cm)** 91.23 ± 12.02	82.48 ± 9.33	87.93 ± 10.43	94.73 ± 12.26	**<0.0001**	**<0.0001**	**<0.0001**	**<0.0001**
**ALT, U/L** 27.73 ± 18.36	20.38 ± 6.57	29.72 ± 17.91	33.83 ± 22.57	**<0.0001**	**<0.0001**	**<0.0001**	**<0.0001**
**AST, U/L** 23.77 ± 8.91	23.42 ± 8.32	24.68 ± 8.15	25.84 ± 10.49	**<0.0001**	**<0.0001**	**0.002**	**<0.0001**
**Total-C, mg/dL** 156.64 ± 30.1	156.94 ± 27.74	153.21 ± 28.50	157.37 ± 31.30	0.33	0.97	0.27	0.44
**LDL, mg/dL** 88.35 ± 33.44	85.65 ± 33.54	85.33 ± 31.64	87.85 ± 34.37	0.62	0.23	0.09	0.23
**HDL, mg/dL** 45.09 ± 10.25	51.08 ± 10.94	51.12 ± 8.55	43.80 ± 10.51	**<0.0001**	0.92	**0.001**	**<0.0001**
**Triglycerides, mg/dL** 95.57 ± 45.80	69.97 ± 24.96	68.99 ± 24.46	103.53 ± 48.72	**<0.0001**	**0.05**	**<0.0001**	**<0.0001**
**Glycaemia, mg/dL** 77.99 ± 8.73	75.79 ± 7.70	77.05 ± 7.15	78.82 ± 9.11	**<0.0001**	**0.008**	**0.001**	**<0.0001**
**Glycemia 2-hr OGTT** 108.05 ± 19.64	103.35 ± 16.44	107.03 ± 16.03	109.89 ± 20.61	**0.001**	**0.003**	**0.006**	**0.001**
**HOMA-IR** 4.30 ± 3.81	1.70 ± 0.82	1.80 ± 1.29	5.23 ± 4.42	**<0.0001**	**<0.0001**	**<0.0001**	**<0.0001**
**Prediabetes, %** 5.2	0	0.7	6.2	**0.001**	0.80	**<0.0001**	**<0.0001**
**PNPLA3 rare allele, carriers %** 51.6	51.9	40.3	58.6	**0.001**	**<0.0001**	**0.001**	**0.001**

Abbreviations*:* ALT, alanine transaminase; AST, aspartate transaminase; BMI, body mass index; DBP, diastolic blood pressure; HDL-C, high-density lipoprotein cholesterol; HOMA, homeostasis model assessment; LDL-C, low density lipoprotein cholesterol; OGTT: Oral glucose tolerance test; PNPLA 3, patatin like phospholipase containing domain 3; SBP: Systolic blood pressure; SDS: Standard deviation score; W/Hr: Waist to height ratio. * *p*1 = comparison between Group 1 and Group 2. ** *p*2 = comparison between Group 2 and Group 3. *** *p*3 = comparison between Group 1 and 3.

## Data Availability

The data presented in this study are available on request from the corresponding author. The data are not publicly available due to the presence of information that could compromise research participant privacy.

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
