# Peer review of "MAFLD in Obese Children: A Challenging Definition"

_children, 2021, doi:10.3390/children8030247_

Round 1

Reviewer 1 Report

The communication is well presented and appropriately raises important questions about how the definition of Metabolic (dysfunction) associated fatty liver disease (NAFLD) can be suitable for obese children.

Minor revisions:

-A flowchart summarizing the criteria for inclusion in the 3 different groups should be added.

-Hepatic steatosis was assessed by liver ultrasound and was evaluated using standard criteria. Ultrasound is the most widely used diagnostic tool to detect steatosis. It is also known that ultrasound has limited sensitivity and it doesn't detect steatosis of <20%, would have been interesting if the authors could discuss the possibility to use a different approach to define NAFLD in overweight/obese children.

Author Response

Reviewer 1

The communication is well presented and appropriately raises important questions about how the definition of Metabolic (dysfunction) associated fatty liver disease (NAFLD) can be suitable for obese children.

Minor revisions:

-A flowchart summarizing the criteria for inclusion in the 3 different groups should be added.

Answer: we added the required flowchart in the revised version of the manuscript. Please see the Figure 1 of the revised version of the manuscript.

-Hepatic steatosis was assessed by liver ultrasound and was evaluated using standard criteria. Ultrasound is the most widely used diagnostic tool to detect steatosis. It is also known that ultrasound has limited sensitivity and it doesn't detect steatosis of <20%, would have been interesting if the authors could discuss the possibility to use a different approach to define NAFLD in overweight/obese children.

Answer: thank you for your valuable comment. We discussed the possibility of different approaches for NAFLD diagnosis in OW/OB children. Please see lines 229-237 of the revised version of the manuscript.

Reviewer 2 Report

 The paper describes the  effectiveness of MAFLD definition in selecting obese children at higher cardiovascular risk. The authors performed clinical, biochemical, and metabolic evaluations  in selecting obese children at higher cardiovascular risk. The results are scientifically valid. The manuscript is well-written and I suggest that, subject to very minor revisions.

line 34: Please expand the NAFLD abbreviation

line 35: There is "evidence of evidence of". Delete one "evidence of".

Author Response

The paper describes the effectiveness of MAFLD definition in selecting obese children at higher cardiovascular risk. The authors performed clinical, biochemical, and metabolic evaluations in selecting obese children at higher cardiovascular risk. The results are scientifically valid. The manuscript is well-written and I suggest that, subject to very minor revisions.

line 34: Please expand the NAFLD abbreviation

Answer: we corrected it accordingly. Please see line 33 of the new version of the manuscript.

line 35: There is "evidence of evidence of". Delete one "evidence of".

Answer: we corrected it accordingly. Please see line 35 of the new version of the manuscript.